# Targeted Affinity Purification and Mechanism of Action of Angiotensin-Converting Enzyme (ACE) Inhibitory Peptides from Sea Cucumber Gonads

**DOI:** 10.3390/md22020090

**Published:** 2024-02-16

**Authors:** Yangduo Wang, Shicheng Chen, Wenzheng Shi, Shuji Liu, Xiaoting Chen, Nan Pan, Xiaoyan Wang, Yongchang Su, Zhiyu Liu

**Affiliations:** 1College of Food Sciences and Technology, Shanghai Ocean University, Shanghai 202206, China; admit1996@126.com (Y.W.); wzshi@shou.edu.cn (W.S.); 2Key Laboratory of Cultivation and High-Value Utilization of Marine Organisms, Fisheries Research Institute of Fujian, Xiamen 361013, China; sjscut@126.com (S.L.); xtchen@jmu.edu.cn (X.C.); npan01@qub.ac.uk (N.P.); 17793478960@163.com (X.W.); 3Medical Laboratory Sciences Program, College of Health and Human Sciences, Northern Illinois University, DeKalb, IL 60015, USA; schen1@niu.edu

**Keywords:** *Apostichopus japonicus*, ACE inhibitory activity, affinity purification, molecular docking, molecular dynamics

## Abstract

Protein hydrolysates from sea cucumber (*Apostichopus japonicus*) gonads are rich in active materials with remarkable angiotensin-converting enzyme (ACE) inhibitory activity. Alcalase was used to hydrolyze sea cucumber gonads, and the hydrolysate was separated by the ultrafiltration membrane to produce a low-molecular-weight peptide component (less than 3 kDa) with good ACE inhibitory activity. The peptide component (less than 3 kDa) was isolated and purified using a combination method of ACE gel affinity chromatography and reverse high-performance liquid chromatography. The purified fractions were identified by liquid chromatography–tandem mass spectrometry (LC–MS/MS), and the resulting products were filtered using structure-based virtual screening (SBVS) to obtain 20 peptides. Of those, three noncompetitive inhibitory peptides (DDQIHIF with an IC_50_ value of 333.5 μmol·L^−1^, HDWWKER with an IC_50_ value of 583.6 μmol·L^−1^, and THDWWKER with an IC_50_ value of 1291.8 μmol·L^−1^) were further investigated based on their favorable pharmacochemical properties and ACE inhibitory activity. Molecular docking studies indicated that the three peptides were entirely enclosed within the ACE protein cavity, improving the overall stability of the complex through interaction forces with the ACE active site. The total free binding energies (ΔG_total_) for DDQIHIF, HDWWKER, and THDWWKER were −21.9 Kcal·mol^−1^, −71.6 Kcal·mol^−1^, and −69.1 Kcal·mol^−1^, respectively. Furthermore, a short-term assay of antihypertensive activity in spontaneously hypertensive rats (SHRs) revealed that HDWWKER could significantly decrease the systolic blood pressure (SBP) of SHRs after intravenous administration. The results showed that based on the better antihypertensive activity of the peptide in SHRs, the feasibility of targeted affinity purification and computer-aided drug discovery (CADD) for the efficient screening and preparation of ACE inhibitory peptide was verified, which provided a new idea of modern drug development method for clinical use.

## 1. Introduction

Angiotensin-converting enzyme inhibitors (ACEIs) regulate blood pressure and maintain cardiac function by suppressing angiotensin II (Ang II) production and deactivating bradykinin through ACE inhibition in human tissues [1]. The development of food-derived ACE inhibitory peptides (ACEIPs) has recently emerged as a highly effective nonpharmacological strategy for preventing and treating hypertension due to its safety, specificity, and efficient absorption by the human body [2]. Due to the varied nature of aquatic environments (differing degrees of salinity, temperature, and light), marine organisms have accumulated many substances with special chemical structures and functional activities in metabolic and biological processes. Ongoing research is focused on the discovery and synthesis of marine compounds through biotechnological approaches, with the goal of increasing the accessibility and chemical diversity of marine-derived functional components. 

In Asian regions, the sea cucumber serves as one of the primary aquaculture products with significant culinary and medicinal importance [3]. ACE inhibitory activity has been previously demonstrated in sea cucumber gonad hydrolysates [4]. However, the large-scale purification of active peptides from marine sources is challenging due to issues related to cost, purity, activity retention, and molecular weight control. The exploration of efficient extraction methods for ACEIPs from sea cucumbers has become a prominent area of research in this field [5]. 

The immobilization of ACE using various immobilization media is a promising affinity purification method to obtain shorter peptides from diverse sources [6]. For instance, Liu et al. successfully utilized a magnetic immobilized metal affinity chromatography matrix modified with poly(ethylene glycol) methyl ether to rapidly purify LLYQEPVLGPVR, a potent ACE inhibitory peptide from casein hydrolysate [7]. Feng et al. used a novel affinity medium Fe_3_O_4_@ZIF-90 and combined this method with reversed-phase high-performance liquid chromatography (RP-HPLC) and MALDI-TOF mass spectrometry to purify the ACE inhibitory peptide KNFL from *Undaria pinnatifida* [8]. Moreover, Lan et al. (2015) employed a novel ACE inhibitory peptide KNFL from magnetically immobilized ACE affinity-purified eluate in conjunction with RP-HPLC to isolate and purify the ACE inhibitory peptide GMKCA [9]. While there has been some progress in recent years, large-scale purification strategies are hindered by the large molecular dimensions of ACE, its susceptibility to inactivation, and high-cost considerations. Consequently, developing affordable immobilized support for ACE is of high importance, especially for the affinity purification of ACE inhibitors.

Computer-aided drug discovery (CADD) is widely employed in drug design and discovery due to its capacity to identify and characterize groundbreaking compounds such as protein inhibitors and antibiotics, thereby achieving significant savings in terms of time and economic costs [10]. CADD has been successfully implemented to predict the pharmacokinetic properties and toxicity (ADMET) of drugs and to conduct virtual screening (SBVS) based on molecular interstructural binding scores. Furthermore, CADD can also identify and validate binding sites in target proteins [11,12]. In this study, molecular docking for ligand–target protein interactions was conducted to elucidate the inhibitor’s mode of action [13]. We also performed a molecular dynamics (MD) simulation and binding free energy analysis to validate these interactions, providing a deeper understanding of the binding process and conformational changes within the complex system. When these methods are combined with the prediction of ADMET properties, they offer a fast and efficient means of screening potential drugs [14]. Finally, the feasibility of this approach was verified by analyzing the in vivo antihypertensive activity in SHRs.

## 2. Results and Discussion

### 2.1. Isolation and Purification of ACE Inhibitory Peptides 

The raw hydrolysates from alcalase treatment (SCGH) (4 mg·mL^−1^) exhibited the best ACE inhibitory activity among the selected protease treatments (Figure 1A), with an inhibition activity of 84.3 ± 0.5% and an IC_50_ value of 589.0 ± 25.0 mg·mL^−1^ (Figure 1C). The peptide components of SCGHs with low molecular weights (<3 kDa) displayed the highest ACE inhibition (Figure 1B), with the lowest IC_50_ value of 303.2 ± 27.0 mg·mL^−1^ (Figure 1C). We further evaluated the molecular weight of SCGH extracts and found that the majority of their components were peptides (98.9%), with a molecular weight of less than 3 kDa. The most abundant peptides had a molecular weight of less than 1 kDa (89.6%) (Figure 1D). By utilizing an immobilized ACE enzyme affinity column (ACE–Sepharose 4B), we successfully obtained two predominant peptide components (Figure 2A). The A2 fraction (1 mg·mL^−1^) exhibited higher ACE inhibitory activity (74.8 ± 0.1%) than those in SCGHs and A1 (62.4 ± 0.2%). HPLC was used for the separation and purification of A2. After excluding the reagent peaks, three components, A2-1, A2-2, and A2-3, were isolated. Among them, the A2-2 fraction (1 mg·mL^−1^) demonstrated the highest ACE inhibitory activity (85.3 ± 0.2%), followed by A2-3 (79.6 ± 0.3%), while the lowest inhibitory activity was observed for A2-1 (70.9 ± 0.5%). 

Collectively, low-molecular-weight fractions isolated from SCGHs using the ultrafiltration method allowed us to obtain small peptides with potent ACE inhibitory activity. ACE inhibitory molecules are typically characterized by their small and short-chain peptide structures [15]. In this study, a combination of ultrafiltration separation and SCGH effectively eliminated substances above 3 kDa. We further isolated the active components by utilizing an ACE enzyme affinity column, which facilitated the removal of other nonspecifically bound substances and yielded the target peptides [16]. The effectiveness of affinity purification and the isolation of active purified fractions were verified by RP-HPLC. Agarose-immobilized ACE affinity chromatography, as shown in this study and others, is a rapid and effective method to purify peptides with high ACE inhibitory activity. For instance, Clan et al. utilized magnetic agarose-immobilized ACE affinity and RP-HPLC methods to isolate the ACE inhibitory peptide GMKCA (IC_50_ = 45.7 μM·L^−1^) [9]. Similarly, Su et al. purified a novel peptide, TLRFALHGME (IC_50_ = 93.5 μM·L^−1^), from *Takifugu flavidus* hydrolysate using a CNBr-activated agarose method [17].

### 2.2. Identification and Pharmacochemical Analysis of ACE Inhibitory Peptides 

The most active subfraction, namely A2-2, was analyzed using LC–MS/MS and identified with the PEAKS Studio software (PEAKS Studio Xpro, Bioinformatics Solutions Inc., Waterloo, QC, Canada). We obtained at least 20 peptides using the virtual screening method SBVS for the binding scoring of the identified peptides (Table 1). These peptides were predicted to be either hydrophilic or amphipathic and composed of 7–12 amino acids, with molecular weights ranging from 872.9 to 1188.2 Da. Furthermore, they appeared to possess traits that facilitate the crossing of the blood–brain barrier (BBB) [18], as evidenced by ADMET properties. Moreover, six peptides, namely HDWWKER, SDDFFNR, ADDFYYQ, LPGDNVGFN, FDDINLH, and DDQYHIF, exhibited favorable absorption properties (30%) during gastrointestinal absorption (HIA) prediction conducted in animals [19]. CYP450 inhibitory potentials were considerably low for all the selected peptides. We anticipate minimal interference with the five major metabolic pathways in drug metabolism, without adverse effects on human health [20]. Additionally, no acute oral toxicity in rats (Ames toxicity or carcinogenicity) was detected in our tests (Table 2).

The 20 leading peptide sequences screened in our studies, utilizing both computational and experimental methods, consist of 7–12 amino acids. The overall hydrophilicity of these peptides may contribute to their ACE inhibitory activity. They can electrostatically interact with the hydrophilic active sites of ACE, influencing their inhibitory effect [21,22]. The six peptides identified through virtual screening exhibit favorable drug-like properties and hold promise for developing ACE inhibitors. SBVS involves molecular docking simulations to assess the binding energies of peptide–protein complexes, while the identified peptides by LC–MS can be further screened by comparing the magnitude of the binding energies [23]. Predicting ADMET properties in peptides can offer supplementary information during the drug development stage, serving as selection criteria and reducing the risk of failure [24]. Hao et al. efficiently identified eight peptides with potential anti-inflammatory activities by employing PeptideRanker in combination with virtual screening based on molecular docking scores [25]. Moreover, Wang et al. identified six peptides with pancreatic lipase inhibitory activity (TF, EW, QWM, NIF, AGY, and PIF) using PeptideRanker, SwissADME, and AutoDock scores [26].

### 2.3. Peptide Synthesis, Inhibition Rate, and Inhibition Type

We synthesized the 20 peptides screened for based on the procedure in Section 2.2 and evaluated their ACE inhibition activities at a concentration of 1 mmol·mL^−1^ (Figure 3A). Among them, DDQYHIF displayed the highest inhibitory activity (66.2 ± 0.9%), followed by HDWWKER (54.0 ± 1.8%), THDWWKER, ADDFYYQ, and SDDFFNR. The effects of different concentrations of the above peptides on IC_50_ were examined. DDQYHIF also exhibited good ACE inhibition, with an IC_50_ value of 333.5 ± 25.3 μmol·L^−1^, followed by HDWWKER, with an IC_50_ value of 583.6 ± 27.6 μmol·L^−1^, consistent with the results in Figure 3A showing the magnitude of ACE inhibitory activity (Figure 3B). It is important to highlight that the octapeptide THDWWKER, which elongates the heptapeptide HDWWKER by adding one threonine at the N-terminus, demonstrated diminished inhibitory activity, indicated by an IC_50_ value of 1291.8 ± 107.6 mmol·L^−1^ (Figure 3C).

The type of inhibition exerted by these inhibitors on the enzyme was initially determined using the Lineweaver–Burk double reciprocal plot method, which involved comparing the Michaelis constant (Km) and the maximum reaction rate (Vmax) [27]. This approach was employed to analyze the enzyme kinetics of DDQYHIF, HDWWKER, and THDWWKER, categorizing the inhibition type of the three peptides (Figure 3D–F). The plots indicate that as the concentrations of the peptides increased, 1/Vmax increased and Vmax decreased, while the Km did not change significantly. This suggests that the mode of inhibition for DDQYHIF, HDWWKER, and THDWWKER may be noncompetitive [28].

Previous studies on the correlation between amino acid composition and ACE inhibition showed that the presence of aromatic amino acids (Trp, Tyr, and Phe) at the C-terminus, as well as hydrophobic (Leu, Ile, Ala, and Met), aromatic (Tyr, Phe, Try, and Thy), and basic amino acids (Lys, Arg, and His) at the N-terminus, favored binding to ACE [29,30,31]. For instance, the experimentally purified short peptide DDQYHIF features Phe at the C-terminus, while HDWWKER has basic amino acids Arg and His at the C- and N-termini, respectively. Sun et al. reported that peptides with Arg at the C-terminus exhibited enhanced ACE activity. Its two novel ACE inhibitors, FGMPLDR and MELVLR, which were prepared in *Ulva intestinalis*, also had low IC_50_ values of 219.4 μM and 236.9 μM, respectively [32]. Zhong et al. (2018) also found that in the preparation of ACE inhibitory peptides from sea cucumber gonads, three novel ACE inhibitory peptides were isolated through gel filtration chromatography and reversed-phase high-performance liquid chromatography, etc., and the resulting IC_50_ values of EIYR and LF were 347.6 μM and 358.4 μM, respectively. We found that by using traditional purification and separation methods, the differences in the ACE inhibitory activity of the prepared peptides and that of DDQYHIF (IC50 = 333.5 ± 25.3) were small [5]. Our results showed that the removal or addition of specific amino acids within the peptide sequence can lead to significant changes in inhibitory activity (HDWWKER vs. THDWWKER) [33]. Compared to traditional separation and purification methods, affinity purification can quickly and efficiently develop peptides with better ACE inhibitory activity.

### 2.4. Molecular Docking Simulation

Molecular docking demonstrated that DDQYHIF, HDWWKER, and THDWWKER were entirely enclosed within the ACE protein cavity. This resulted in a favorable alignment with the center of the active pocket and augmented the overall stability of the complex through intermolecular interaction forces (Figure 3). DDQYHIF forms hydrogen bonds with several residues, including Gln281, Tyr523, Arg522, Ala354, Arg402, Tyr394, and Tyr360. It also forms π–π stacking interactions with Hie410, Phe512, and Phe527. Salt bridges were formed with Glu403 and Lys511, and hydrophobic interactions were observed with Phe457, Tyr523, Ala354, Hie410, Val380, Hid383, and Tyr360. Similarly, HDWWKER forms hydrogen bonds with Gln281, Hid383, Hie353, Ala354, Ala356, Asp358, Lys511, Tyr520, and Tyr523. It also forms π–π stacking interactions with Hie410 and Tyr360. Salt bridges are formed with Asp358, Glu376, Asp415, Asp453, and Lys511, and hydrophobic interactions occur with Phe512, Hie410, Pro407, Tyr523, Phe457, Val380, and Val351. THDWWKER forms hydrogen bonds with Gln281, Tyr523, Tyr520, Arg522, Ala354, Hie353, Ala356, Asn66, and Asp358. It also forms a π–π stacking interaction with Hie410. Salt bridges are formed with Asp415, Hie353, and Lys511. A π–cation interaction occurs with Phe359, and hydrophobic interactions are observed with Phe457, Phe512, Tyr523, Ala354, Hie410, Glu403, and Tyr62, where His in DDQYHIF, Trp in HDWWKER and Lys in THDWWKER all exhibited metal coordination with Zn701. 

ACE possesses three active pockets, namely S1, S2, and S1′. The S1 pocket is characterized by the presence of amino acid residues Ala354, Glu384, and Tyr523, while the S2 pocket contains Gln281, His353, Lys511, His513, and Tyr520. In contrast, Glu162 was identified in the S1′ pocket [34]. The coenzyme factor, with Zn^2+^ as the active center site of ACE, can bind to amino acid residues His383, His387, and Glu411, thereby forming a tetrahedral structure that increases the stability of the complex [35]. Table 3 shows that THDWWKER, HDWWKER, and DDQYHIF can interact with amino acid residues in the active pockets S1 and S2 but not with the residues in the active pocket S1′, which is similar to lisinopril. This may partially explain why these peptides have significant ACE activity. Among these, only HDWWKER and DDQYHIF interacted with His383, which possibly forms the tetrahedron in the active center. This interaction could lead to the stable tetrahedral structural distortion of this residue with Zn^2+^, resulting in a subsequent decrease in ACE catalytic activity [36]. As illustrated in Figure 4B,C, the deletion of the first Thr residue in THDWWKER disrupts the previously stable hydrogen (hydrogen bonding interaction) and ionic bonds (π–cation interaction) with Asp358 and Phe359 residues, thereby altering its original steric conformation. In contrast, the overall structure of the HDWWKER is more extended, exposing functional areas (Lys structural region) that would otherwise be buried while moving closer to the ACE active site, allowing for more interactions (hydrogen bonding with Lys and the tetrahedral amino acid active residue His383). These factors likely contribute to the higher ACE inhibitory activity observed in HDWWKER. However, further simulation analysis and verification of the interaction relationship are necessary from the perspective of dynamics and binding ability size.

### 2.5. Molecular Dynamics Simulation

The root-mean-square deviation (RMSD) analysis indicated stability at 6 ns for the ACE–DDQYHIF complex and 10 ns for both the ACE–HDWWKER and ACE–THDWWKER complexes. The RMSD values for these complexes consistently hovered around 1.7 Å, 1.8 Å, and 1.9 Å, respectively, exhibiting a fluctuation amplitude consistently constrained to less than 0.3 Å. This shows that the three systems reached equilibrium during the simulation and could be used for later computational analysis [37]. A comparative analysis of the radius of gyration (Rg) profiles for the ACE–peptide complexes and the ACE monomer revealed similar patterns of fluctuation, with the Rg values stabilizing post-simulation at 23.8 (Figure 5B). In the three ACE–peptide complex systems, the locations of the peaks with higher root-mean-square fluctuation (RMSF) values are analogous, encompassing α-helical structures within the protein’s lateral head region (Asp40-Ala45 and Val104-Gln108), with additional peaks constituted by loop structures (Figure 5C–F). These peak areas are situated on the outer surface of the protein, exposed to the solvent, thereby endowing the residues with greater flexibility and, consequently, elevated RMSF values.

The blue-shaded areas in Figure 5C–E highlight the peptide–protein binding regions. Within these regions, the average RMSF values for the unbound ACE protein are typically higher than the bound states in the complexes [37]. The binding site residues, comprising α-helices, β-sheets, and loop structures, are situated near the active pocket within the protein’s cavity. Notably, in the ACE–HDWWKER complex, the RMSF values for the loop region Val350-Tyr360 at the binding site are higher than those observed in the ACE monomer. The conformational difference maps for each complex, relative to the monomeric protein, are presented in Figure 5G [38]. In the three ACE–peptide complex systems, residues in the head peak region undergo significant displacement, and the solvent-exposed loop structures shift to varying degrees away from the center. Internally, the residues within the binding site area tend to shift toward the active center. However, in the ACE–HDWWKER complex, the loop region Val350-Tyr360, involved in hydrogen bonding, undergoes bond disruption [39].

### 2.6. Binding Free Energy Calculations

The total binding free energy (ΔG_total_) of the ACE–DQYHIF complex system was determined to be −21.9 Kcal·mol^−1^ (Table 4). The van der Waals contribution (ΔG_vdw_) was found to be −79.8 Kcal·mol^−1^, indicating that the stronger interhydrophobic interactions play a dominant role in the binding process. The higher values observed for electrostatic forces can be attributed to the preference for disrupted electrostatic interactions over the newly formed ones during the binding of the ligand and protein [40]. This preference resulted in an unfavorable effect, leading to positive values for the electrostatic forces. The hydrogen bond interaction involving the active residue His383 may lead to the formation of a stable tetrahedral structural distortion with Zn^2+^. The polar solvation energy (ΔG_polar_) counteracted the contribution of nonpolar solvation energy (ΔG_nonpolar_) to the binding energy, resulting in a solvation energy (ΔG_solv_) value of only −0.7 Kcal·mol^−1^. 

The ACE–HDWWKER and ACE–THDWWKER complex systems demonstrated similar compositions in terms of decomposition values. They also exhibited total free energy differences (ΔG_total_) of −71.6 Kcal·mol^−1^ and −69.1 Kcal·mol^−1^, respectively. Electrostatic interactions played a crucial role in the binding process, as indicated by ΔG_ele_ values of −411.8 Kcal·mol^−1^ and −444.1 Kcal·mol^−1^, respectively. Furthermore, interhydrophobic interactions moderately contributed to the binding, resulting in higher ΔG_gas_ values of −505.5 Kcal·mol^−1^ and −529.4 Kcal·mol^−1^. These findings provide further supporting evidence for the binding mechanism of the peptide–protein interactions and suggest that all three peptides can bind to ACE rapidly, forming stable and favorable complex structures [41].

### 2.7. Antihypertensive Activity

The spontaneously hypertensive rat (SHR) is a widely used animal model for studying human essential hypertension [17]. After the intravenous administration of THDWWKER, HDWWKER, and DDQIHIF, antihypertensive effects were evaluated in SHRs. As shown in Figure 6, HDWWKER had good effects on lowering blood pressure. A significant reduction in SBP was observed between 2 and 8 h (*p* < 0.05), with the lowest SBP of 176 mmHg occurring around 4 h after HDWWKER was administered. The SBP then recovered to 220 mmHg after 12 h. 

The relationship between in vitro ACE inhibition and antihypertensive activity is not obvious due to complex biological factors such as degradation by digestive enzymes, the intestinal barrier, and plasma peptidases; therefore, it is particularly important to study the hypotensive effects of ACE-inhibiting peptides in vivo [42]. HDWWKER is a novel ACE inhibitory peptide that not only showed potent ACE inhibitory activity in vitro but also exhibited potent and sustained antihypertensive effects in SHRs. This result implies that HDWWKER not only has good drug chemistry and high ACE inhibitory activity in vitro but also maintains high inhibitory activity in vivo, and thus it has great potential as an antihypertensive drug.

## 3. Materials and Methods

### 3.1. Materials

Sea cucumber gonads were purchased from Ningde City (Fujian, China). All mature sea cucumber specimens were transported through a complete cold chain to the laboratory, where they were stored at −20 °C to ensure their quality was not compromised. Concentrated hydrochloric acid, methanol, boric acid, borax, etc., were purchased from Sinopharm Group (Beijing, China). Alkaline protease was purchased from PangBo Bioengineering Co., Ltd. (Nanning, China). Hippuryl-His-Leu (HHL), angiotensin-I-converting enzyme (ACE), hippuric acid (HA), and CNBr-activated Sepharose 4B, acetonitrile (CAN) and trifluoroacetic acid (TFA) were purchased from Sigma-Aldrich (St. Louis, MO, USA). All other chemicals/reagents used were of analytical grade or HPLC grade.

### 3.2. Preparation of Peptides from Sea Cucumber Gonads

Fresh sea cucumber gonads were lyophilized using a vacuum freeze-dryer (model Alpha2-4LDplus, Christ, Osterode, Germany) before being ground into powder. The optimum protease was selected based on ACE inhibitory activity from alkaline protease, neutral protease, and pepsin. The sea cucumber gonad proteolytic solution was then subjected to ultrafiltration separation. Initially, the hydrolysate was filtered through a machine equipped with a 200 nm ceramic membrane (model CeraMem-0100, Xiamen Fumex Technology Co., Ltd., Xiamen, China) to remove large particulate impurities and bacteria. Subsequently, the solution was further fractionated using ultrafiltration centrifugal tubes (3 KDa and 10 KDa) (UFC800396 Millipore Amicon, Shanghai Bitter Biotech Co., Ltd., Shanghai, China) to obtain different molecular weight fractions (<3 KDa, 3–10 KDa, and >10 KDa).

### 3.3. Assay of ACE Inhibitory Activity 

The ACE inhibitory activity of the samples was determined using a high-performance liquid chromatograph (HPLC) (model Waters e2695, Alliance, San Diego, CA, USA) equipped with a SunFire C18 column (4.6 mm × 150 mm; Merck, Darmstadt, Germany), as previously described by Su et al. (2023) [17].

### 3.4. Determination of Molecular Weight of Hydrolysate

The molecular weight (MW) distribution of sea cucumber gonadal hydrolysate was determined and analyzed using a high-performance liquid chromatograph equipped with a TSKgel G2000 SWXL gel column (300 mm × 7.8 mm; Tosoh Co., Ltd., Tokyo, Japan) coupled with Empower GPC processing software (Empower 3, Agilent, Palo Alto, CA, USA) [23]. The chromatograms of sea cucumber gonad hydrolysates were obtained at a flow rate of 1 mL/min, an injection volume of 10 μL, a UV detection wavelength of 228 nm, and a mobile phase ratio of acetonitrile–water (containing 0.1% trifluoroacetic acid) of 45:55 and analyzed with the Empower GPC processing software. Cytochrome C (Mw 12,384 Da), peptidase (Mw 6512 Da), bacillus subtilisin (Mw 1423 Da), Gly-Gly-Tyr-Arg (Mw 451 Da), and Gly-Gly-Gly (Mw 189 Da) were used for calibration, and standardized fitting curves and third-order regression models were plotted.

### 3.5. Targeted Affinity Purification 

An activated gel made from cyanogen bromide-activated agarose 4B packing (Seplite 4B) was chelated with the solubilized ACE protein–ligand. The ACE-immobilized gel packing was obtained by using 0.1 M acetate buffer (containing 0.5 M NaCl) at pH 4.0 and Tris-HCl buffer (containing 0.5 M NaCl) at pH 8.0. This column is an immobilized enzyme gel targeted affinity adsorption column. Pure, small-molecule peptides from sea cucumber gonads served as the upper sample solution for targeted affinity adsorption. The peptide was dissolved in borate buffer (50 mmol·L^−1^, pH 7.4) and loaded onto the immobilization ACE affinity column pre-equilibrated with 50 mmol·L^−1^ borate buffer (pH 7.4). After loading, the column was washed with equilibration buffer (50 mmol·L^−1^ borate, pH 7.4) until the absorbance at 220 nm returned to a stable baseline. The elution was performed using a borate buffer with a concentration of 0.1 M (containing 1 M NaCl) at pH 8.3, as the mobile phase, with a flow rate of 0.6 mL·min^−1^. The UV220 absorbance peaks were collected, desalted, and lyophilized for the next experiments.

### 3.6. Analysis and Purification by RP-HPLC 

Using the same instrument configuration as in Section 3.3, the hydrolysate before and after ultrafiltration and the target affinity eluent were subjected to analysis using RP-HPLC and verified for activity through gradient elution. The samples were filtered using a 0.22 μm organic-phase microporous filter membrane and loaded onto the sample at a flow rate of 0.5 mL·min^−1^. The mobile phase A consisted of water containing 0.1% TFA, while phase B comprised acetonitrile containing 0.1% TFA. An elution gradient was applied, ranging from 5% to 30% of phase B for 30 min [9].

### 3.7. Peptide Sequence Identification by LC–MS 

The elution fractions of the targeted affinity were subjected to desalting and analyzed using a Q-Exactive mass spectrometer (Thermo Fisher, Waltham, MA, USA) equipped with an Acclaim PepMap C18 column (75 μm × 25 cm, Thermo Fisher, Waltham, MA, USA) and an online nanospray ion source. The mobile phase A consisted of water containing 0.1% formic acid, while phase B was composed of 80% acetonitrile containing 0.1% formic acid. The elution gradient ranged from 2% to 35% of phase B over a period of 0–47 min. This was followed by a rapid increase to 100% phase B within 1 min, which was then maintained for 12 min.

The parameters for the mass spectrometry analysis were set as follows: mass/charge (*m*/*z*) = 200–2000. For the primary MS analysis, the resolution was set at 70,000, the AGC target was 3 × 10^6^, and the maximum ion trap (IT) was 50 ms. For the secondary MS/MS analysis, the resolution was 17,500, the topN was 20, the isolation window was 2 *m*/*z*, the AGC target was 1 × 10^5^, the maximum IT was 45 ms, and the NCE/Stepped NCE was 28 kV. A dynamic exclusion time of 30 s was applied. The mass spectral raw files were analyzed using Peaks Studio version 10.6 (Bioinformatics Solutions Inc., Waterloo, ON, Canada).

### 3.8. Computer-Assisted Virtual Screening of Peptides 

Peptides with a −10lgP value greater than 25.0 and a peptide chain length less than 20 were screened using −10lgP as a confidence indicator. Peptides that were found to have ACE inhibitory activity were excluded by utilizing the online databases BIOPEP-UWM and AHTPDB (http://crdd.osdd.net/raghava/ahtpdb/ (accessed on 3 June 2023)). The unidentified active peptides obtained from the screening process were then subjected to activity prediction using the online system PeptideRanker (http://bioware.ucd.ie/~compass/biowareweb/Server_pages/peptideranker.php (accessed on 5 June 2023)), and only peptides with a bioactivity score (PR) of 0.5 or higher were considered. The ACE protein was selected as the receptor, and the ACE protein crystal structure (PDB ID: 1O8A) was retrieved from the RCSB protein database (http://www.rcsb.org (accessed on 3 June 2023)). The ACE molecules were preprocessed using the docking software UCSF DOCK 6.9 (UCSF, University of California, San Francisco, CA, USA). This involved several steps, including hydrogenation, charge filling, water removal, and protonation. The Zn^2+^ ion was retained while distant Cl- was removed, and energy minimization was performed using the Amber ff12SB treatment. Small-molecule peptides were mapped using Discovery Studio2019 Client (BIOVIA Inc., California, CA, USA), which were then used as ligands to finalize the active pocket of ACE with Zn^2+^ as the active center, defining the active coordinates as (X:43.11, Y:39.404, Z:47.919), with a pocket range of 10 and a box edge of 6. The DOCK 6.9 program was used to carry out virtual screening based on molecular docking. The top 20 peptides with the highest scores, as determined by the Grid Score ranking of the virtual screening results, were selected for solid-phase synthesis.

### 3.9. Peptide Synthesis, Medicinal Chemical Properties, and Activity Screening

The 20 peptides with the highest scores were synthesized by Kingsley Biotechnology Ltd. (Nanjing, China) and yielded a purity greater than 98%. The molecular weights and isoelectric points were determined using Expasy’s pl/Mw online tool (http://web.expasy.org/compute_pi/ (accessed on 8 June 2023)). The peptides’ Grand average of hydropathicity (GRAVY) and toxicity were computed using ToxinPred (https://webs.iiitd.edu.in/raghava/toxinpred/ (accessed on 8 June 2023)) [43]. The pharmacokinetic properties of the peptides were analyzed using the online tool ADMETlab (https://admetmesh.scbdd.com/ (accessed on 9 June 2023)), focusing on indicators such as human intestinal absorption (HIA), blood–brain barrier (BBB) penetration, CYP450 metabolism inhibition parameters, and acute oral toxicity [14]. The ACE inhibitory activities of the 20 peptides were determined at a concentration of 1 mg/mL, and the peptides with superior activities were selected. The ACE inhibition rates for the selected peptides were examined at different concentrations, and their IC50 values were calculated to verify their effectiveness.

### 3.10. Inhibitory Kinetic Study 

The inhibition kinetics of DDQYHIF, HDWWKER, and THDWWKER were analyzed using Lineweaver–Burk plots according to the protocol described by Lin et al. (2017) [44]. The mixture for the DDQYHIF reaction consisted of 100 μL HHL (3, 4, 5, and 6 mmol·L^−1^) as substrate, 50 μL of 5 mmol·L^−1^ACE, and 100 μL of the sample solution (0.1, 0.5, and 1 mmol·L^−1^ DDQYHIF). The mixture of HDWWKER and THDWWKER reactions consisted of 100 μL of HHL (2, 3, 4, 5, and 6 mmol·L^−1^) as substrate, 50 μL of 5 mmolL ^−1^ ACE, and 100 μL of sample solution (0.1, 0.5, and 1.5 mmol·L^−1^ HDEEKER; 0.2, 1, and 2 mmol·L^−1^ THDWWKER). Inhibition was determined under the same conditions as in Section 3.3. Lineweaver–Burk plots were plotted as the reciprocal of the initial reaction rate (1/V) versus substrate concentration (1/[S]), and Km and Vmax were calculated.

### 3.11. Molecular Docking 

The semi-flexible molecular docking of peptides with improved ACE inhibitory activity was conducted using the DOCK 6.9 program. The ACE protein receptor and ACE inhibitory peptide were treated as described in Section 3.8, with the protein receptor set as rigid and the peptide ligand set as flexible. Default parameters were used for other settings. The active center sites of the docking box were set to coordinates (X:43.11, Y:39.404, Z:47.919) Å, and the size of the box was constrained to dimensions (21.918, 25.707, 28.752) Å. The grid module was employed to generate scoring and evaluation grid points within the box, and semi-flexible docking was performed to generate 10,000 different conformations. Cluster analysis was carried out with a root-mean-square deviation (RMSD) queue value of 2.0 Å. The final output consisted of the top-scoring conformations from the 10,000 different orientations. Intermolecular interactions such as hydrogen bonding, hydrophobic interactions, and salt bridges were analyzed in the best conformations using the PyMoL 2.5 molecular visualization system (Delano Scientific LLC, San Carlos, CA, USA). 

### 3.12. Molecular Dynamics (MD) Simulation 

Molecular dynamics (MD) simulations were carried out using AmberTools 20 [45] (AMBER 2020, University of California, San Francisco, CA, USA). AMBER ff19SB [46] and GAFF [47] force fields were used for proteins and compounds, respectively. The system was solvated by a truncated octahedron water box using the OPC water model with a margin of 10 Å. Periodic boundary condition (PBC) was used, and the net charge was neutralized with 0.15 M of NaCl. Nonbonded van der Waals interactions were calculated using the Lennard–Jones 12-6 potentials with a 10 Å cutoff, while long-range electrostatics were treated using the particle mesh Ewald (PME) algorithm. The SHAKE [48] algorithm was applied to constrain bonds involving hydrogen atoms. To remove improper atom contacts, the structure was first minimized by (1) 5000 steps of steepest descent and 5000 steps of conjugate gradient, under a harmonic constraint of 10.0 kcal/(mol·Å2) on heavy atoms; and (2) relaxing the entire system by 5000 steps of steepest descent and 5000 steps of conjugate gradient, and then gradually heating the system to 300 K by a 20 ps NVT simulation. Two equilibration phases were then conducted: (1) a 0.02 ns NPT simulation with constraints on heavy atoms, followed by (2) a 1 ns NVT simulation without restraints. The temperature was maintained at 300 K using the Berendsen thermostat with a 1 ps coupling constant, and the pressure was set at 1 atm using the Monte Carlo barostat with a 1 ps relaxation time. Finally, the system underwent a 20 ns NVT simulation with a time step of 2 fs. The root-mean-square deviation (RMSD), root-mean-square fluctuation (RMSF), radius of gyration (Rg), and conformational differences were analyzed using the CPPTRAJ [49] module.

### 3.13. Binding Free Energy Calculations

The binding free energies were calculated using the molecular mechanics Poisson–Boltzmann surface area (MM/PBSA) [50] method implemented in AmberTools 20 for 200 snapshots from the MD trajectory. For each snapshot, the free energy was calculated for the receptor, the ligand, and the complex using a “single-trajectory” approach. The total binding free energy was calculated according to the following equation:$$\Delta G_{bind}=\Delta G_{complex}-(\Delta G_{receptor}+\Delta G_{ligand})=\Delta E_{vdw}+\Delta E_{ele}+\Delta G_{PB}+\Delta G_{SA}-T\Delta S$$
where $\Delta G_{complex}$, $\Delta G_{receptor}$, and $\Delta G_{ligand}$ represent the free energy of the complex, protein, and ligand; $\Delta E_{vdw}$, $\Delta E_{ele}$, $\Delta G_{PB}$, and $\Delta G_{SA}$ refer to the van der Waals energy, electrostatic energy, and the polar and nonpolar components of the desolvation-free energy. TΔS represents the conformational entropy contribution at temperature *T*. The $\Delta G_{PB}$ term was determined using the PB model, while $\Delta G_{SA}$ was evaluated as $\Delta G_{SA}=\gamma \times SASA+\beta$, where SASA was the solvent accessible surface area and the values of the constants γ and β were 0.00542 kcal·Å^−^^2^ and 0.92 kcal·mol^−^^1^, respectively. The molecular surface was determined using solvent probes with a radius of 1.4 Å.

### 3.14. Antihypertensive Effect In Vivo

Male SHRs (10 weeks, 230 ± 20 g body weight) were purchased from Vital River Laboratory Animal Technology Co., Ltd. (Beijing, China). Following an acclimation period of one week, SHRs with systolic blood pressure (SBP) higher than 220 mmHg were randomly divided into six animals per group. Animals were housed at 25 °C under a light–dark cycle of 12 h. THDWWKER, HDWWKER, and DDQYHIF were injected via the tail vein at a dose of 12 mg/kg body weight.

Under the same conditions, a saline group was used as the model control and captopril (5 mg·kg^−1^) as the positive control. The SBP of all rats was measured using an ALC-NIBP blood pressure monitor (Alcott Biotech, Shanghai, China) before and after intravenous administration. The rats were all treated humanely according to the guidelines of the National Institutes of Health and Use of Laboratory Animals and approved by the Ethics Committee of Guangdong Medical Laboratory Animal Center (no. 20211001, approved on 9 September 2021).

### 3.15. Statistical Analysis

The data from each sample group were assessed for significant differences using SPSS 22.0 software (SPSS Institute, Cary, NC, USA) and then graphed using Origin 2023 software (OriginLab, Northampton, MA, USA).

## 4. Conclusions

In this study, enzymatic hydrolysis using alkaline proteases followed by ultrafiltration yielded components with molecular weights of less than 3 kDa (SCGHs), which exhibited the highest ACE inhibitory activity. The completeness of the ultrafiltration was confirmed by molecular weight distribution analysis, with most substances being less than 1 kDa. Peptides with potent ACE inhibitory activity were purified from SCGHs using immobilized ACE affinity chromatography (ACE–Sepharose 4B column) combined with reverse-phase high-performance liquid chromatography (RP-HPLC). This method effectively enriched ACE inhibitory peptides and removed a significant amount of nontarget peptides. The purified fractions were identified by LC–MS/MS, and the identified peptides were selected using SBVS to obtain peptides with good ACE inhibitory activity. Three novel ACE inhibitory peptides (DDQIHIF, HDWWKER, and THDWWKER) with favorable drug-like properties were discovered, exhibiting IC_50_ values for the ACE inhibition of 333.5 μmol·L^−1^, 583.6 μmol·L^−1^, and 1291.8 μmol·L^−1^, respectively. Enzyme kinetic studies indicated that the mode of inhibition for all three peptides was noncompetitive. Molecular docking, dynamics, and binding free energy analysis confirmed that these peptides could bind rapidly and favorably to the active site of ACE, resulting in a tighter and more stable complex after binding. A short-term assay of antihypertensive activity in spontaneously hypertensive rats (SHRs) revealed that HDWWKER could significantly decrease the systolic blood pressure (SBP) of SHRs after intravenous administration. Therefore, based on the validation of in vivo antihypertensive activity in SHRs, targeted affinity purification combined with computer-aided drug discovery (CADD) can rapidly, efficiently, and accurately predict ACE inhibitory peptides with in vivo and in vitro antihypertensive activity, providing a new approach for the discovery of bioactive peptides of food origin. Further implications of these methods lie in the development of drugs for the treatment of hypertension.

## Figures and Tables

**Figure 1 marinedrugs-22-00090-f001:**
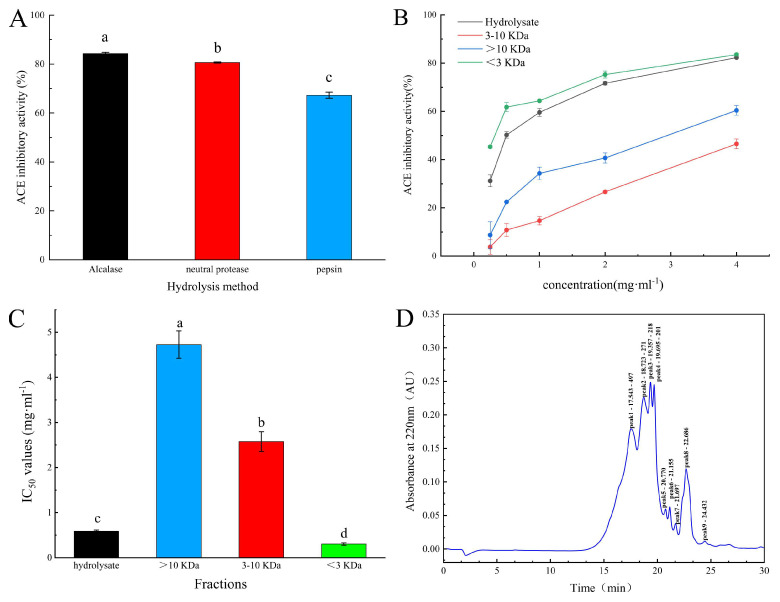
ACE inhibitor activity and molecular weight distribution of ultrafiltration hydrolysates: (**A**) ACE inhibitory activity assay of different proteases hydrolysates; (**B**) the ACE-inhibiting activity of the ultrafiltration fractions was assessed at various levels of concentration; (**C**) IC_50_ values for ACE inhibitory activity of the ultrafiltration fractions; (**D**) determination of molecular weight distribution of SCGH using gel permeation chromatography (TSK G2000 gel chromatography columns). Different lowercase letters (a–d) on the bars mean that the difference is significant (*p* < 0.05).

**Figure 2 marinedrugs-22-00090-f002:**
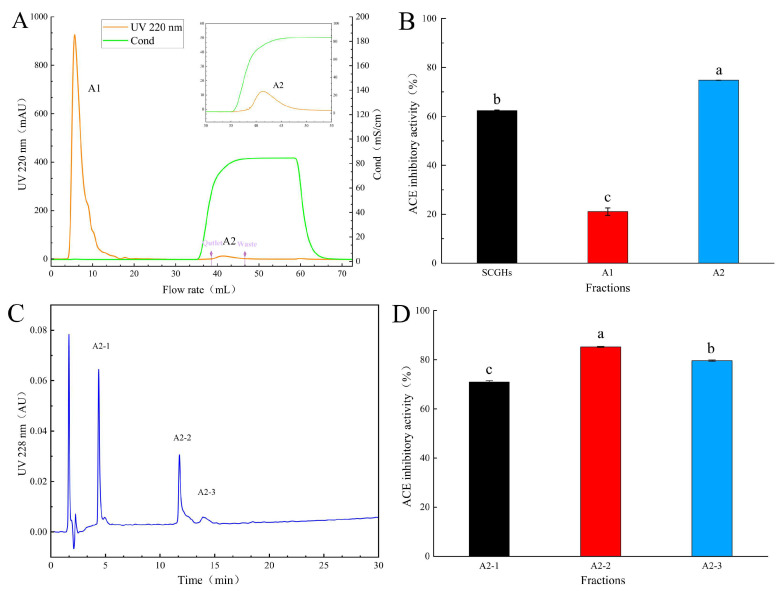
Purification, column separation, and enzyme adsorption of peptides: (**A**) ACE–Sepharose 4B column affinity adsorption plot; (**B**) ACE inhibitory activity of different affinity adsorption components; (**C**) RP-HPLC separation of the targeted affinity eluent (C18 column); (**D**) ACE inhibitory activity of the purified component by RP-HPLC. Different lowercase letters (a–c) on the bars mean that the difference is significant (*p* < 0.05).

**Figure 3 marinedrugs-22-00090-f003:**
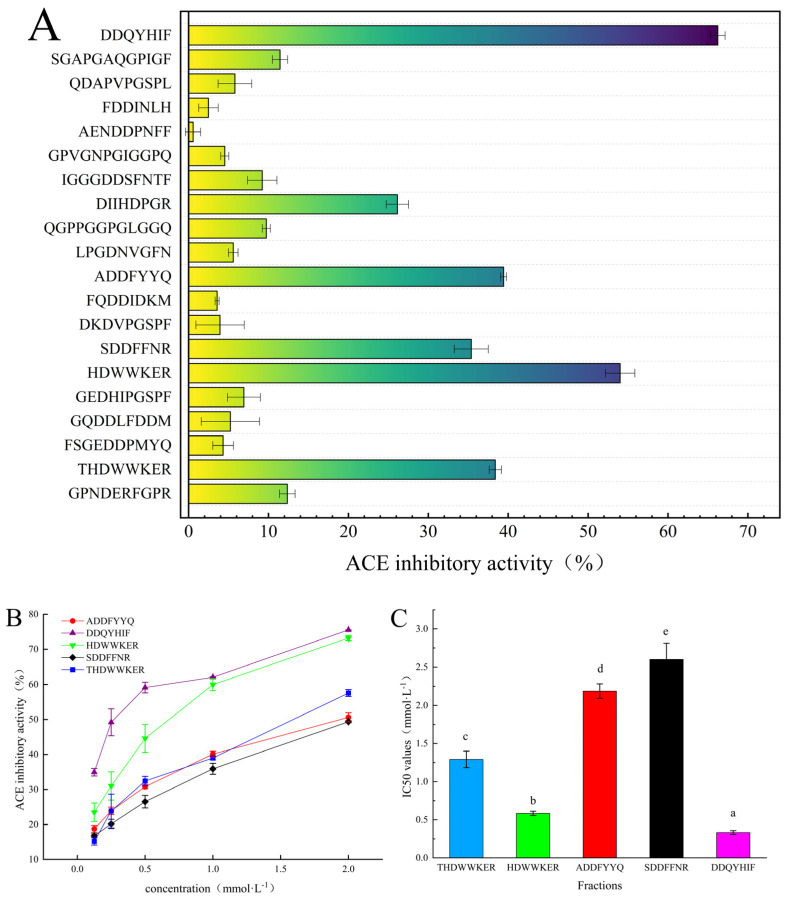
Inhibition rate and inhibition type: (**A**) ACE inhibitory activity was measured for each peptide at a concentration of 1 mg/Ml; (**B**) ACE inhibitory activity of each peptide at different concentrations; (**C**) IC_50_ values of ACE inhibitory activity of each peptide; (**D**) the Lineweaver–Burk plots of the reactions of ACE in the presence of DDQIHIF; (**E**) the Lineweaver–Burk plots of the reactions of ACE in the presence of HDWWKER; (**F**) the Lineweaver–Burk plots of the reactions of ACE in the presence of THDWWKER. [S] = hippuryl-l-histidyl-l-leucine concentration; V = velocity of the reaction. Different lowercase letters (a–e) on the bars mean that the difference is significant (*p* < 0.05).

**Figure 4 marinedrugs-22-00090-f004:**
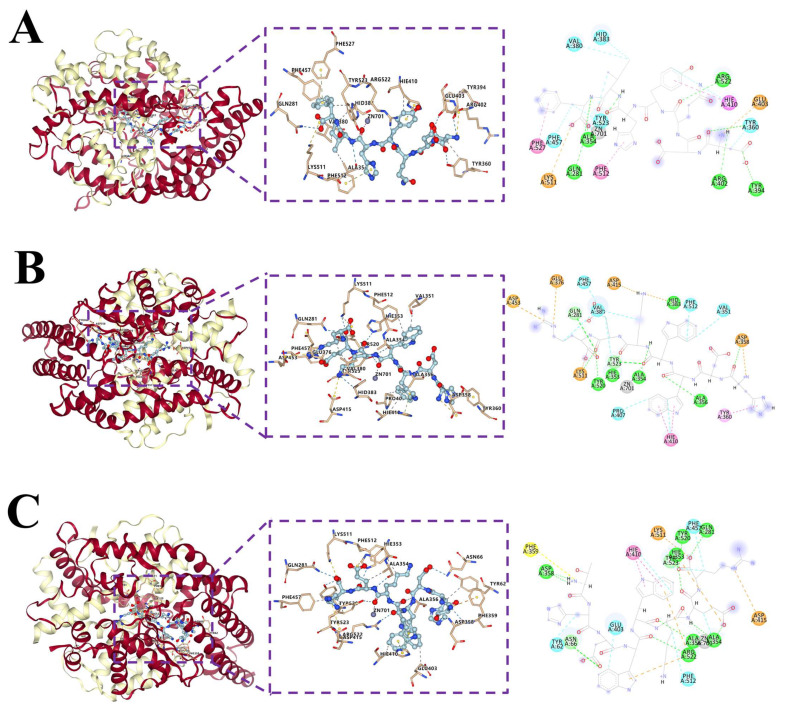
The 3D and 2D plots of molecular docking of peptides with ACE: (**A**) the ACE–DDQYHIF complex; (**B**) the ACE–HDWWKER complex; (**C**) the ACE–THDWWKER complex. The chart displays different interactions: salt bridge forces (indicated by the orange dotted line), hydrogen bond forces (indicated by the dashed blue line), hydrophobic interaction forces (indicated by the gray dashed line), π–π stacking forces (indicated by the green dashed line), zinc ion forces (indicated by the purple dashed line), and π–cation interactions (indicated by the yellow dashed lines). In the protonation of the receptor, histidine is protonated under different pH conditions and is HID when the hydrogen atom is at the δ-position nitrogen atom and HIE at the ε-position nitrogen atom, i.e., His513, His353, His383, His387, and His410 are transformed to Hie513, Hie353, Hid383, Hid387, and Hie410.

**Figure 5 marinedrugs-22-00090-f005:**
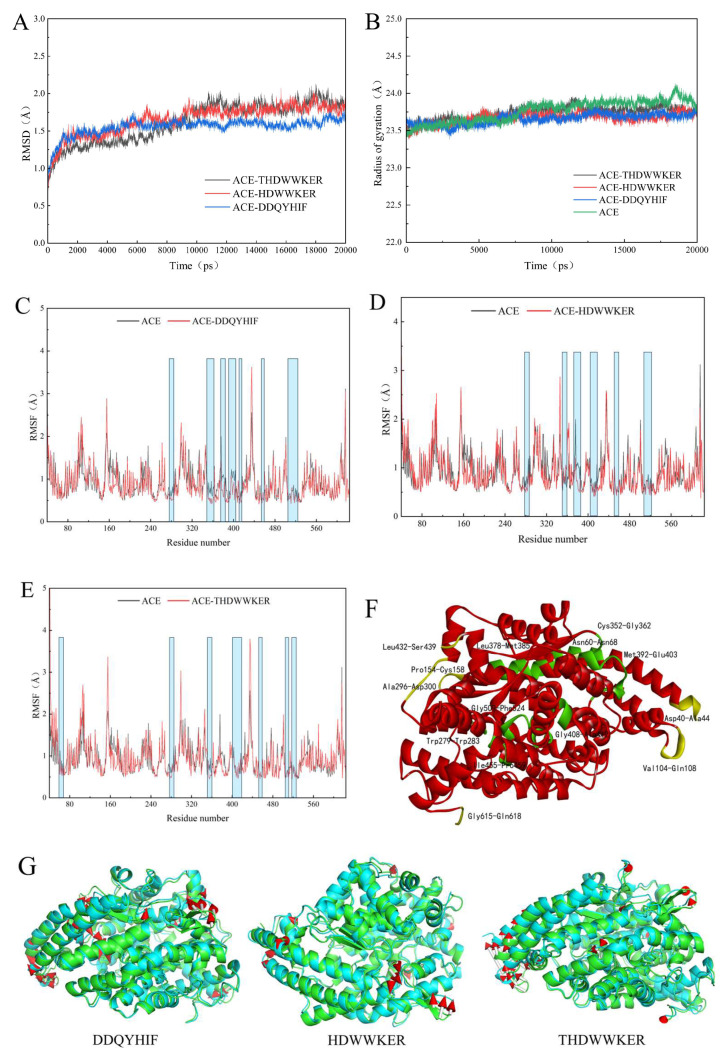
Molecular dynamics (MD) simulation: (**A**) RMSD curve of ACE–peptide complex system; (**B**) Rg curve of ACE–peptide complex system; (**C**) RMSF curve of ACE–DDQYHIF complex system; (**D**) RMSF curve of ACE–HDWWKER complex system; (**E**) RMSF curve of ACE–THDWWKER complex system; (**F**) ACE protein 3D map; (**G**) diagram of conformational differences between ACE–peptide complex systems and protein monomers. Different color cartoons in the figure represent different systems: the green cartoon is the ACE monomer protein, the blue cartoon is the ACE–peptide complex system, and the arrow points from the ACE monomer protein to the ACE–peptide complex system. The length of the arrow indicates the size of the difference between the two systems.

**Figure 6 marinedrugs-22-00090-f006:**
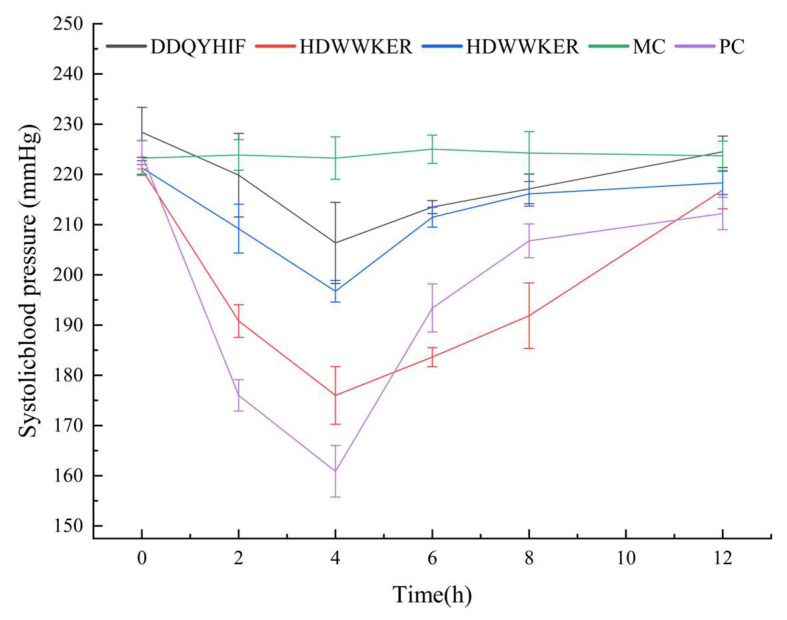
SBP changes in SHRs after intravenous administration. MC indicates model control (intravenous 0.9% saline). PC indicates positive control (intravenous 5 mg·kg^−1^ captopril).

**Table 1 marinedrugs-22-00090-t001:** Virtual screening and analysis of physicochemical properties of ACE inhibitory peptides.

No	Peptide	Mass (Da)	Water Solubility	Isoelectric Point	GRAVY	−10lgP	Peptide Ranker	Grid Score (kJ/mol.)
1	GPNDERFGPR	1144.21	Insoluble	6.07	−2.07	34.50	0.814	−164.7258
2	THDWWKER	1157.25	Soluble	6.42	−2.64	29.53	0.554	−155.6739
3	FSGEDDPMYQ	1188.23	Soluble	3.49	−1.34	32.13	0.531	−144.8246
4	GQDDLFDDM	1055.08	Insoluble	3.32	−1.04	39.68	0.653	−144.1646
5	GEDHIPGSPF	1055.11	Soluble	4.35	−0.77	29.06	0.667	−140.5179
6	HDWWKER	1056.15	Soluble	6.75	−2.91	28.1	0.735	−139.6188
7	SDDFFNR	899.92	Soluble	4.21	−1.46	28.21	0.881	−138.9027
8	DKDVPGSPF	961.04	Soluble	4.21	−0.92	26.55	0.526	−137.0551
9	FQDDIDKM	1011.11	Insoluble	3.93	−1.09	27.61	0.582	−134.7892
10	ADDFYYQ	920.93	Insoluble	3.56	−1.21	26	0.612	−134.1931
11	LPGDNVGFN	931.99	Soluble	3.8	−0.23	33.28	0.652	−133.1769
12	QGPPGGPGLGGQ	1021.09	Soluble	5.52	−0.87	33.74	0.828	−132.4893
13	DIIHDPGR	922.01	Soluble	5.21	−0.96	31.59	0.517	−130.6882
14	IGGGDDSFNTF	1129.15	Insoluble	3.56	−0.28	42.29	0.607	−130.2659
15	GPVGNPGIGGPQ	1049.15	Insoluble	5.52	−0.42	45.69	0.625	−129.655
16	AENDDPNFF	1068.06	Insoluble	3.49	−1.3	42.99	0.746	−126.6069
17	FDDINLH	872.93	Insoluble	4.2	−0.37	25.82	0.511	−125.8939
18	QDAPVPGSPL	980.09	Soluble	3.8	−0.32	35.46	0.670	−125.2897
19	SGAPGAQGPIGF	1058.16	Soluble	5.24	0.15	38.45	0.867	−124.7403
20	DDQYHIF	936.98	Insoluble	4.2	−1.1	30.33	0.673	−124.5179

Note: GRAVY indicates the grand average of hydropathy; −10lgP is the confidence indicator.

**Table 2 marinedrugs-22-00090-t002:** Analysis of ADMET properties of ACE inhibitory peptides.

No	Peptide	ADMET					
		HIA	BBB	CYP450 Inhibits Confounding	Rat Oral Acute Toxicity	AMES Toxicity	Carcinogenicity
1	GPNDERFGPR	-	+	Low	-	-	-
2	THDWWKER	-	+	Low	-	-	-
3	FSGEDDPMYQ	-	+	Low	-	-	-
4	GQDDLFDDM	-	+	Low	-	-	-
5	GEDHIPGSPF	-	+	Low	-	-	-
6	HDWWKER	+	+	Low	-	-	-
7	SDDFFNR	+	+	Low	-	-	-
8	DKDVPGSPF	-	+	Low	-	-	-
9	FQDDIDKM	-	+	Low	-	-	-
10	ADDFYYQ	+	+	Low	-	-	-
11	LPGDNVGFN	+	+	Low	-	-	-
12	QGPPGGPGLGGQ	-	+	Low	-	-	-
13	DIIHDPGR	-	+	Low	-	-	-
14	IGGGDDSFNTF	-	+	Low	-	-	-
15	GPVGNPGIGGPQ	-	+	Low	-	-	-
16	AENDDPNFF	-	+	Low	-	-	-
17	FDDINLH	+	+	Low	-	-	-
18	QDAPVPGSPL	-	+	Low	-	-	-
19	SGAPGAQGPIGF	-	+	Low	-	-	-
20	DDQYHIF	+	+	Low	-	-	-

Note: + indicates significance; - indicates no significance; HIA indicates human intestinal absorption; BBB indicates Blood–brain barrier.

**Table 3 marinedrugs-22-00090-t003:** Interaction of lisinopril, an inhibitory peptide with amino acid residues in the active pocket of ACE.

Active Pocket	Amino Acid Residue	Lisinopril	THDWWKER	DDQYHIF	HDWWKER
S1	Ala354	+	+	+	+
	Tyr523	+	+	+	+
	Glu384	+	-	-	-
S2	Gln281	-	+	+	+
	His353	-	+	-	+
	Lys511	-	+	+	+
	His513	+	-	-	-
	Tyr520	+	+	-	+
S1′	Glu162	-	-	-	-
T	His383	+	-	+	+
	His387	+	-	-	-
	Glu411	+	-	-	-

Note: + indicates the presence of an interaction with an amino acid residue; - indicates no mutual interaction with amino acid residues. Captopril is a synthetic nonpeptide angiotensin-converting enzyme (ACEI) inhibitor that acts primarily on the renin–angiotensin–aldosterone system (RAAS system). It can prevent the conversion of angiotensin I or angiotensin II and inhibit aldosterone secretion, reducing sodium retention.

**Table 4 marinedrugs-22-00090-t004:** Predicted binding free energies of ACE–peptides complexes.

Energy Term	ACE–DDQYHIF	ACE–HDWWKER	ACE–THDWWKER
ΔG_vdw_	−79.8 ± 5.1	−93.7 ± 7.5	−85.2 ± 6.3
ΔG_ele_	58.6 ± 16.9	−411.8 ± 40.5	−444.1 ± 21.9
ΔG_polar_	7.6 ± 17.6	443.5 ± 33.8	470.5 ± 22.2
ΔG_nonpolar_	−8.4 ± 0.3	−9.6 ± 0.2	−10.2 ± 0.3
ΔG_gas_	−21.2 ± 17.3	−505.5 ± 38.3	−529.4 ± 20.1
ΔG_solv_	−0.7 ± 17.4	433.9 ± 33.8	460.3 ± 22.1
ΔG_total_	−21.9 ± 9.4	−71.6 ± 11.4	−69.1 ± 11.7

Note: ΔG_vdw_, ΔG_ele_, ΔG_polar_, ΔG_nonpolar_, ΔG_solv_, and ΔG_gas_ are binding energy components of van der Waals, electrostatic, polar, nonpolar, solvation, and gas energies, respectively.

## Data Availability

The data presented in this study are available on request from the corresponding author.

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
