# Peer review of "Targeted Affinity Purification and Mechanism of Action of Angiotensin-Converting Enzyme (ACE) Inhibitory Peptides from Sea Cucumber Gonads"

_marinedrugs, 2024, doi:10.3390/md22020090_

Round 1
Reviewer 1 Report
Comments and Suggestions for Authors
This appears to be a thorough investigation of ACE inhibition by sea cucumber gonad extracts. Unfortunately, the paper suffers from a lack of flow, with different sections interleaved. There is no discussion or conclusion. The RESULTS section 2.1-2.6 leads straight into the METHODS section, which is also numbered 2. I suspect that METHOD originally came before RESULTS and when the paper was rearranged the numbering was not corrected. On line 448 reference is made to method 3.3.
There is no comparison of the results presented with results in the literature. Reference is made to previous studies, but no actual numbers are given. How do these IC50values compare? How do the binding energies compare? The significance of the work is difficult to judge as there is nothing to compare it with. If these peptides are worse than previously studied peptides from other organisms, then it is not significant.
The explanation of the peptide docking to the active site is confusing. In the free monomer what is the Zn coordinated to? Do the peptides coordinate to the Zn?
Throughout the manuscript numbers are given to 3-4 decimal places. This is totally unrealistic and makes one wonder if the authors appreciate what their numbers mean. For computational results I would only quote whole numbers and for IC501 decimal place at most.
Figure 1 is confusing. The legend refers to the preparation and molecular weight determination, but the figure is actually about ACE inhibitory activity. Figure 1A has three bars labelled ‘a’ b’ and ‘c’. Are these the same as the bars ‘a’, ‘b’, ‘c’, and ‘d’ in Figure 1C, because they do not appear to be. Figure 1A x-axis has ‘concentration (4 mg ml-1)’ but in fact the variable is hydrolysis method. Also how was the concentration of 4 mg ml-1 calculated? Figure 1D has an insert that is never referred to.
Figure 2B. should read ‘ACE inhibitory activity of different affinity adsorption components’. Figure 2C has 3 peaks labeled (the first is not, why?), A2-1, A2-2 and A2-3. Figure 2D has 3 bars labelled A2-a, A2-b and A2c??? Again at the top of the bars there are ‘a’, ‘b’ and ‘c’???
Line 123 states that 20 peptides were obtained by SBVS. In the experimental it says that 20 peptides were found in the low MW fraction of the eluent. Which is it?
Line 129 refersto gastrointestinal absorption (HIA) tests conducted in animals, but there is no reference to this, and it is not described in the experimental section. I think the confusion may be with the word ‘test’, which I take to mean a physical test rather than a calculated parameter.
On line 123 reference is made to SBVS. Then on line 139, SBVS is explained? Surely that sentence should read: We obtained at least 20 peptides using SBVS, a virtual screening tool (Table 1). SBVS, uses molecular docking simulations to assess the binding energies of peptide-protein complexes to identify peptides with specific activities [21]. While this method is powerful, its application is limited mainly due to computational power and time constraints.
Table 1 and 2. Legend abbreviations need to be explained.
Line 157. Again the 20 peptides were identified by virtual screening and not in the hydrolysate?
L161 Effects of different concentration of the above peptides on IC50 was examined. Surely IC50 value is point at which have 50% activity. How then can you change the concentration?
Figure 3B. Why are the results consistent with those obtained at 1 mM concentration? Line 163
Figure 3 Delete ‘Peptide synthesis’
Amino acid codes Hie and Hid are not standard. Please define. I assume it refers to the protonation state of His but it is not a standard definition.
Paragraphs starting line 194 and 212 are out of order.
Line 219 “active pockets S1 and S2, but not with the residues in the active pocket S1”. This must be a mistake.
Line 210. Which atom of the 3 peptides coordinates to the Zn?
Line 225. “deletion of the first Thr residue in THDWWKER disrupts the previously stable hydrogen and ionic bonds with Asp358 and Phe359 residues”. To what are Asp358 and Phe359 bonded to that Thr disrupts?
Line 227. In contrast, HDWWKER has a greater extent, and its functional regions are exposed. What does “extent” mean and what are the functional regions exposed to?
Figure 4. I cannot see the interactions with Zn.
Table 3, use of lisinopril is not explained.
Figure 5G, different colours are not explained.
Binding free energy calculation. I have problems with this as the numbers are so large that when they are subtracted the difference is very uncertain. How meaningful can 443(+-33) minus 411(+-40) be? The difference is less than the standard deviation!
Line 316 “A total of 20 peptides were identified through LC-MS/MS analysis” but in the text it is said they were identified by SBVS.
The whole manuscript needs careful editing.
Comments on the Quality of English Language
See comments above
Reviewer 2 Report
Comments and Suggestions for Authors
Comments and suggestions to authors:
Targeted affinity purification and mechanism of action of angiotensin-converting enzyme (ACE) inhibitory peptides from sea cucumber gonad
Line 40. complete the phrase "hold special significance". What is the special significance?
Line 42-44. I consider that the wording of these lines should be improved. I could start "With the aim of improving...” end "investigations have been focused…"
Line 56. Correctly write condensed formulas, with subscripts
Line 93-96. Mention the figures where these results are observed
Line 107-111. Mentioned the results obtained from other similar investigations; however, it is necessary to compare the results obtained in this investigation with the references that are mentioned. Do they obtain a better IC50?, are the results they obtained higher or lower? Etc.
Line 126. missing bibliographic citation
Line 129. missing bibliographic citation
Line 141-142. Eliminate “While this method is powerful, its application is limited mainly due to computational power and time constraints”
Line 175. Missing bibliographic citation
Line 220. In the development of the article, lisinopril and its action in ACE are not mentioned.
Line 221. “Only HDWWKER and DDQYHIF interacted with His383, which possibly forms the tetrahedron in the active center”, In what figure is this behavior observed?
Line 226. Thereby altering its original steric conformation. Why is this phenomenon important?
Line 230. These factors likely contribute to the higher ACE inhibitory activity observed in 230 HDWWKER. Compared to lisinopril, is it better or worse? (more discussion).
Line 245-246. The Root Mean Square Deviation (RMSD) analysis indicates stability at 6 ns for the 245 ACE-DDQYHIF complex and at 10 ns for both the ACE-HDWWKER and ACE- 246 THDWWKER complexes. What do these results mean?
Line 247-257. The explanation of results is good, however, there is a lack of comparison with other studies to improve the paper.
Line 305. Missed bibliographic cited
Line 308-327. It remains to conclude on the importance of the use of sea cucumber gonad and relate it to the results obtained.
References
The way to put the references is not homogeneous, please check.
Example
[16]. Tehseen, M.; Raducanu, V.; Rashid, F. Proliferating cell nuclear antigen-agarose column: A tag-free and tag-dependent tool for 554 protein purification affinity chromatography. J Chromatogr A. 2019, 1602, 341-349. 555
[17]. Su, Y.; Chen, S.; Liu, S. Affinity Purification and Molecular Characterization of Angiotensin-Converting Enzyme (ACE)- 556 Inhibitory Peptides from Takifugu flavidus. Mar Drugs 2023, 21, 522.
Round 2
Reviewer 1 Report
Comments and Suggestions for Authors
The authors have addressed most of my comments. The English could still be improved. I still do not understand the use of letters on the bar graphs to indicate statistical significance.
Comments on the Quality of English Language
The authors have addressed most of my comments. The English could still be improved. I still do not understand the use of letters on the bar graphs to indicate statistical significance.
Author Response
Response to Reviewer 1 Comments
Point 1: The English could still be improved.
Response: We have made further changes to some of the words and grammar in the text.
Point 2: still do not understand the use of letters on the bar graphs to indicate statistical significance.
Response: Line 118, 125, 213, We modify the corresponding statement as follows: “Different lowercase letters (a–d) on the bars mean that the difference is significant (p < 0.05).”
Reviewer 2 Report
Comments and Suggestions for Authors
After carefully reviewing this document, I consider that the document it is better. However, in the PDF document I make some observations such as improving some figures, in addition to the fact that in Materials and Methods there are still several references missing.
I attach the PDF because in it I made some observations
